



# Least travel time ray tracer, version Two (LTT v2) adapted to the grid geometry of the OpenIFS atmospheric model

Maksym Vasiuta[1], Angel Navarro Trastoy[1], Sanam Motlaghzadeh[1], Lauri Tuppi[1], Torsten Mayer-Gürr[2], and Heikki Järvinen[1]

[1]Institute for Atmospheric and Earth System Research / Physics, Faculty of Science, University of Helsinki, Finland
[2]Technical University of Graz, Institute of Geodesy, Graz, Austria

**Correspondence:** Maksym Vasiuta (maksym.vasiuta@helsinki.fi)

**Abstract.** Electromagnetic signals commonly used in geodetical applications, such as the Global Navigation Satellite System (GNSS), undergo bending and delay in the neutral gas atmosphere of the Earth. The least-travel-time (LTT) concept is one of the approaches to model signal slant delays via a ray tracing (RT) procedure. In this study, we developed an LTT-based RT algorithm (LTT v2), where the 3-dimensional refractivity field of the atmosphere is based on the atmospheric model data.

This representation is complete in a sense that the domain of the RT conforms to the native grid geometry of the atmospheric model. In addition, some physical and numerical approximations are improved compared to the previous version (LTT v1). The atmospheric states are generated using a global numerical weather prediction model, the Open Integrated Forecast System of the European Centre for Medium-Range Weather Forecasts. The slant delays of LTT v2 are compared with the products of the original least-travel-time GNSS delay model (LTT v1) and the products of the state-of-the-art VieVS Ray Tracer (RADIATE).

The skill of slant delay estimation is assessed using metrics that are indicative of the quality of GNSS products derived using the GROOPS (Gravity Recovery Object Oriented Programming System) orbit solver software toolkit of the Graz University of Technology. Employment of slant delay products of the LTT RT algorithm shows radical improvement in GNSS processing. When using LTT v2 delay estimates, the GNSS orbit midnight discontinuities are reduced by more than 10% compared to RADIATE, and more than 2% compared to LTT v1. The residuals of ground station precise positioning are analysed with

respect to the IGS14 reference positions. The RMS of residuals (accuracy) and standard deviation (precision) are substantialy reduced compared to RADIATE.

## 1 Introduction

The neutral gas atmosphere of the Earth delays propagation of electromagnetic waves. The delay develops due to wave group velocity being less than in the vacuum, i.e., the speed of light. In addition, the inhomogeneity of air invokes the bending of a

wavefront from a straight line. Thus, the time delay of an electromagnetic signal in the Earth's atmosphere consists of delay along the path and excess of the path length due to bending, both of which are often measured in length units. In space geodesy, the measured distances are thus larger than the geometrical distances.





Signals of the Global Navigation Satellite System (GNSS) are affected by the neutral gas atmosphere (a.k.a. 'tropospheric') delay. The tropospheric delay is one of the major error sources in GNSS processing (Landskron and Böhm, 2018; Wilgan et al., 2017). The state-of-the-art modelling setup, described in IERS Conventions (Petit and Luzum, 2010), represents the tropospheric delay between a surface site and a satellite as an empirical mapping function with adjustable parameters (Böhm et al., 2006): zenith wet and hydrostatic delays (ZWD, ZHD), mapping coefficients (providing elevation dependence) and so-called tropospheric gradients (providing azimuth dependence). The *a priori* mapping parameters are estimated using ray tracing (Landskron and Böhm, 2018; Hofmeister and Böhm, 2017). Ray tracing (RT) is a direct approach to computing the delay by solving the signal propagation through given atmospheric conditions in a specified direction. Quality of the RT procedure is therefore of great importance for obtaining accurate GNSS products. Various signal RT methods are available for geodetical applications. Hofmeister (2016) developed the RADIATE ray tracer for signals of the microwave and the optical frequency ranges, based on the Eikonal equation. Another example is the Least-Travel Time (LTT) ray tracer by Eresmaa et al. (2008). It is based on Snell's Law equation expressed in a polar coordinate system, as explained by Rodgers (2000). These slant delay models are designed differently, and are simplified in one way or another by assumptions in favour of efficiency and utility.

A signal RT algorithm must be supplied with a 3-dimensional representation of the atmospheric refraction index. Generally, the microwave atmospheric refraction index is a function of pressure, temperature and relative humidity (Bevis et al., 1994). The atmospheric state is usually provided by numerical weather prediction models as 2-dimensional or 3-dimensional gridded fields. Numerical weather prediction (NWP) systems, whose data can be employed for signal RT, are continuously improving their spatial and temporal representation of flow dynamics and physical processes as well as advancing the use of Earth observations in data assimilation (Bauer et al., 2015). In the data assimilation perspective, an RT algorithm can be seen as an operator from model state space (model variables) to observation space (delays of signals traversing the atmosphere). This way, the advances in both NWP and signal delay modelling are interconnected. In fact, use of signal delays in NWP data assimilation requires an accurate RT model (Järvinen et al., 2007).

With improvements in NWP models' skill and steadily increasing accuracy of precise orbit determination in GNSS, it is necessary to also revise models of atmospheric signal delay. A crucial area of improvement of an RT algorithm is ensuring lossless utilization of atmospheric information. With this motivation, we revised the LTT RT model and its physical assumptions. We developed a new software with a reworked LTT ray tracer at its core that has more robust physical assumptions: the LTT version Two (LTT v2). This software implementation is designed to optimally utilize NWP data produced by the Open Integrated Forecasting System model (OpenIFS) of the European Centre for Medium-Range Weather Forecasts (ECMWF). The software can be modified to other NWP data input, if needed.

The new LTT v2 model (Vasiuta, 2024) is compared with two other models which are capable of using the OpenIFS input data. The first reference is the LTT code by Eresmaa et al. (2021), which is a modification of the original LTT implementation by Eresmaa et al. (2008). This model is refered to as 'LTT v1'. The second reference is RADIATE ray tracer. Given its distinct structure, a careful adjustment of the OpenIFS data is necessary before input into RADIATE. To ensure a fair comparison, a specific configuration is established for RADIATE, providing the ray tracer with the same atmospheric states and aligning its output structure to match that of LTT v2.





We test the quality of the slant delay models by solving the GNSS precise orbit determination (POD) using a direct coupling of POD and NWP models (Navarro Trastoy et al., 2022). We compute orbit and station position solutions based on observations

of Global Positioning System (GPS) constellation in December 2016 using GPS multi-frequency measurements from 205 ground stations. The GNSS processing is performed with GROOPS toolkit (Mayer-Guerr et al., 2021), which is modified to enable various representations of tropospheric delay estimates. The quality of GNSS products is assessed using two sets of metrics: orbit midnight discontinuity (MD) and precise point positioning (PPP) deviations. The MD metric evaluates the quality of batch orbit estimation, which inherently has a disconnection of two consecutive batch POD solutions both in position

and velocity (Montenbruck and Gill, 2012). We assess positional MD, not the velocity. The PPP-related metrics are root-mean-square deviation and standard deviation from a reference station position. These PPP metrics inform of the accuracy of solutions and show the consistency of the PPP procedure, respectively. This analysis is utilized to determine the relative skill of the three ray tracers: LTT v1, LTT v2, and RADIATE.

In Section 2, we explain the state-of-the-art concept of slant delay modelling via RT. In Section 3, we describe our method-

ology of slant delay modelling, as well as the features of LTT v2. Section 4, explains skill assessment of the three slant delay algorithms via GNSS processing. The slant delay product comparison and skill assessment are reported in the Section 5. Finally, Section 6 attempts to attribute the skill of slant delay modelling to different factors.

## 2  Background

### 2.1  Atmospheric ray tracing methods

The classical approach to ray tracing (RT) a signal in a refractive medium, such as the Earth's atmosphere, is by using the so-called Eikonal equation. The Eikonal equation (Hamilton, 1828) describes an electromagnetic wave propagating through a slowly varying medium, connecting physical (wave) optics and geometric (ray) optics. It reads

$$\|\boldsymbol{\nabla} L\|^2 = n(\boldsymbol{r})^2 \tag{1}$$

where $L$ is the optical path length, $\boldsymbol{\nabla} L$ the components of ray direction, $n$ the known refractive index of the medium, and $\boldsymbol{r}$

the position vector. The surfaces of equal optical length ($L(\boldsymbol{r}) = const$) are called geometrical wave surfaces or geometrical wavefronts.

This equation is usually described in the Hamiltonian canonical formalism (Born et al., 1999; Nilsson et al., 2013):

$$H(\boldsymbol{r}, \boldsymbol{\nabla} L) \equiv \frac{1}{a} \left\{ (\boldsymbol{\nabla} L \cdot \boldsymbol{\nabla} L)^{\frac{a}{2}} - n(\boldsymbol{r})^a \right\} = 0, \tag{2a}$$

$$\frac{\mathrm{d}\boldsymbol{r}}{\mathrm{d}u} = \frac{\partial H}{\partial \boldsymbol{\nabla} L}, \tag{2b}$$

$$\frac{\mathrm{d}\boldsymbol{\nabla} L}{\mathrm{d}u} = -\frac{\partial H}{\partial \boldsymbol{r}}, \tag{2c}$$

$$\frac{\mathrm{d}L}{\mathrm{d}u} = \boldsymbol{\nabla} L \cdot \frac{\partial H}{\partial \boldsymbol{\nabla} L}. \tag{2d}$$





where $a$ is a value related to the parameter of interest $u$, and $H(\boldsymbol{r}, \boldsymbol{\nabla} L)$ is the Hamiltonian. When applying RT to the computation of delay along the path in the atmosphere, the parameter $u$ is the length $s$ along the ray, $a$ equals to 1, and a spherical coordinate system is used (Hofmeister, 2016; Nafisi et al., 2012; Böhm and Schuh, 2013). The formulation of the tropospheric delay problem is thus the following:

$$H\left(r, \vartheta, \lambda, L_r, L_\vartheta, L_\lambda\right) \equiv \left(L_r^2 + \frac{1}{r^2}L_\vartheta^2 + \frac{1}{r^2\sin^2\vartheta}L_\lambda^2\right)^{\frac{1}{2}} - n(r, \vartheta, \lambda, t) = 0, \tag{3a}$$

$$\frac{\mathrm{d}r}{\mathrm{d}s} = \frac{1}{\omega}L_r, \tag{3b}$$

$$\frac{\mathrm{d}\vartheta}{\mathrm{d}s} = \frac{1}{\omega}\frac{L_\vartheta}{r^2}, \tag{3c}$$

$$\frac{\mathrm{d}\lambda}{\mathrm{d}s} = \frac{1}{\omega}\frac{L_\lambda}{r^2\sin^2\vartheta}, \tag{3d}$$

$$\frac{\mathrm{d}L_r}{\mathrm{d}s} = \frac{\partial n(r, \vartheta, \lambda, t)}{\partial r} + \frac{1}{\omega r}\left(\frac{L_\vartheta^2}{r^2} + \frac{L_\lambda^2}{r^2\sin^2\vartheta}\right), \tag{3e}$$

$$\frac{\mathrm{d}L_\vartheta}{\mathrm{d}s} = \frac{\partial n(r, \vartheta, \lambda, t)}{\partial\vartheta} + \frac{1}{\omega}\frac{L_\lambda^2}{r^2\sin^3\vartheta}, \tag{3f}$$

$$\frac{\mathrm{d}L_\lambda}{\mathrm{d}s} = \frac{\partial n(r, \vartheta, \lambda, t)}{\partial\lambda}, \tag{3g}$$

$$d = L - G = \int_S n(r, \vartheta, \lambda, t)\mathrm{d}s - G \tag{3h}$$

where $\mathrm{d}s$ is the path element, $r$ the radial distance, $\vartheta$ the co-latitude ($0 \le \vartheta \le \pi$), $\lambda$ the longitude ($0 \le \lambda \le 2\pi$), and $\omega$ is equal to refractive index $n(r, \vartheta, \lambda, t)$. The ray propagation is solved by integrating Eqs. 3b - 3g w.r.t. $(\boldsymbol{r}, \boldsymbol{\nabla} L)$ starting from some initial value $(\boldsymbol{r_0}, \boldsymbol{\nabla} L_0)$. The delay $d$ is obtained by subtracting geometrical distance $G$ from optical path length $L$ along the path $S$, Eq. 3h.

To solve Eqs. 3b - 3g in general 3-dimensional case, one must develop a rule to compute gradients of the refractive index $n$, which is a challenging task in practice. Furthermore, the computational complexity of a strict approach surges with an increase in zenith angle and atmospheric data resolution. Thus, many practical ray tracer implementations benefit in computation and design by simplifying refractive index gradient effects and adding other assumptions. For example, three-dimensional RT can be reduced to two-dimensional RT by setting horizontal gradient components of the refractive index to zero: $\partial n/\partial\vartheta = 0$ and $\partial n/\partial\lambda = 0$. In this way, the ray propagation is limited to a plane of fixed azimuth.

Analytical approach of Thayer (1967) further simplifies the computation by assuming the refractive index is only dependent on height, and refractivity between two neighbouring layers to be approximated using a power-law relation. In this scenario, the refractive index has only one vertical profile, hence RT constrains slant delays to have azimuthal symmetry. Another 2D RT approach is the so-called 'piecewise-linear ray tracing' explained by Hobiger et al. (2008) and Hofmeister (2016). In this approach, the ray is propagated according to simple Snell's law, where the ray bends only at the half-height layer. Horizontal refractivity gradient is also neglected here, implying azimuthal asymmetry since refractivities on the vertical layers are obtained by bi-linear interpolation from a horizontal grid. Both approaches are employed in the RT program RADIATE, with the 'piecewise-linear' method being the default option.





An alternative design of atmospheric signal RT was proposed by Rodgers (2000). In their approach, Rodgers (2000) considered the Eikonal equation (1) in a two-dimensional coordinate system, instead of restricting the general 3D case. From the 2D Eikonal equation, the law of ray propagation was derived as follows

$$\frac{\mathrm{d}\epsilon}{\mathrm{d}s} = \frac{1}{n(s,t)}\frac{\partial n(s,t)}{\partial t} \tag{4}$$

where $\epsilon$ is the direction of propagation, and s, t the coordinates along and perpendicular to the ray respectively. By expressing Eq. 4 in polar coordinates, and introducing angle $\theta$ between the ray direction and the radius vector, the system of differential equations is obtained, as follows:

$$\frac{\mathrm{d}r}{\mathrm{d}s} = \cos\theta \tag{5a}$$

$$\frac{\mathrm{d}\psi}{\mathrm{d}s} = \frac{\sin\theta}{r} \tag{5b}$$

$$\frac{\mathrm{d}\theta}{\mathrm{d}s} = -\sin\theta\left(\frac{1}{r} + \frac{\partial n(r,\psi)}{\partial r}\right) + \frac{\cos\theta}{n(r,\psi)\cdot r}\frac{\partial n(r,\psi)}{\partial\psi} \tag{5c}$$

Here $ds$ is a path element, $r$ the distance from the Earth's centre of curvature (a.k.a., the Euler radius of curvature), $\psi$ the horizontal counterpart for polar coordinates, $\theta$ the local zenith angle, and $n$ the refractive index. Equations 5 are integrated numerically starting from the ground receiver position ($r_{rec}$; $\psi_{rec} = 0$) in the initial direction defined as $\theta_0$. Since $\theta_0$ is initially unknown, finding a signal path between ground receiver and satellite, satisfying Eqs. 5, is an initial value problem. The resulting tropospheric slant delay $d$ is expressed as follows

$$d = \int_S n(r,\psi)\mathrm{d}s - G = \int_S (n(r,\psi) - 1)\mathrm{d}s + \left(\int_S \mathrm{d}s - G\right) \tag{6}$$

Equation 6 explicitly shows two components of the delay: delay along the path, and the excess of the path length due to bending, or so-called geometrical delay.

## 2.2 The least travel time algorithm, version One

The Least Travel Time (LTT) algorithm by Eresmaa et al. (2021) is based on the theory of Rodgers (2000). In LTT, the starting direction is set as the geometrical zenith angle of the satellite ($\theta_0 = \theta_g$). The integration ends at the satellite altitude ($r_{end} = r_{sat}$). Yet, due to bending, an angular separation appears between the endpoint of the ray and the satellite location, i.e., $r_{end} = r_{sat}$ but $\psi_{end} \neq \psi_{sat}$. Therefore, the ray propagation is repeated by using an updated $\theta_0'$ as follows:

$$\theta_0' = \theta_0 - (\psi_{end} - \psi_{sat}) = \theta_0 - \Delta\psi \tag{7}$$

The final slant delay is approximated as a linear combination of the two delay estimates with different starting zenith angles following Eqs. 6 and 7.

Several assumptions are made in the LTT v1. First, the ambiguity of the initial zenith angle $\theta_0$ is not solved strictly. Also, the geometrical delay term in Eq. 6 is neglected. Moreover, only the radial refractive index gradient is present ($\frac{\partial n(r,\psi)}{\partial\psi}$ is zero in





Eq. 5c). These assumptions make LTT v1 computationally efficient and convenient to use in data assimilation of GNSS signal
delays in NWP models employing the LTT operator's adjoint counterpart (Eresmaa and Järvinen, 2006; Järvinen et al., 2007).
However, they add nonphysical uncertainty to the slant delay products.

### 2.3    Coordinate system and gravity

The ray tracing (RT) of a signal in the atmosphere deals with geometrical variables (lengths, angles). Hence, for the RT

procedure, a reference coordinate system, which is consistent with the geometry of the Earth, should be selected. On the
other hand, atmospheric data from numerical weather models is typically based on a coordinate system consistent with the
Earth's gravity. To use the information from NWP models in RT applications, it is necessary to bring station position data and
atmospheric model grid to the same height system. The ground station height is usually expressed with respect to a reference
ellipsoid, while the heights in NWP models are w.r.t. a geopotential. Transformations for a position at a geographic location

$(\phi, \lambda)$ between geopotential height and orthometric height, as well as orthometric height and ellipsoidal height are as follows
(Hobiger et al., 2008):

$$H = \frac{\Phi}{\overline{\gamma}(\phi, \lambda, H)} = \frac{\zeta \cdot g_0}{\overline{\gamma}(\phi, \lambda, H)} \tag{8a}$$

$$h = H + N(\phi, \lambda) \tag{8b}$$

where $\Phi$ is the geopotential value, $g_0$ the standard acceleration due to gravity, $\zeta$ the geopotential height, $\overline{\gamma}(\phi, \lambda, H)$ the mean

acceleration due to gravity, $H$ the orthometric height, $N$ the geoid undulation, and $h$ the ellipsoidal height.

### 2.4    Refractivity

The refractive index of air $n$ is commonly expressed with refractivity value $N$ as shown in Eq. 9a. Under the ideal gas assump-
tion, refraction of microwave signal follows the relationship (Eq. 9b), being a function of dry air and water vapour pressures
($p_d$ and $e$) and temperature $T$ (Bevis et al., 1994). Modifying the relationship for $N$ to be a function of pressure $p$, temperature

$T$ and specific humidity $q$, which are the common atmospheric variables in NWP models, the new formula was proposed by
Eresmaa and Järvinen (2006) leading to Eq. 9c.

$$n = 1 + 10^{-6} \cdot N \tag{9a}$$

$$N = k_1 \frac{p_d}{T} + k_2 \frac{e}{T} + k_3 \frac{e}{T^2} \tag{9b}$$

$$N = \frac{k_1 p}{T} + \frac{(k_2 - k_1)qp}{(\epsilon + (1-\epsilon)q)T} + \frac{k_3 qp}{(\epsilon + (1-\epsilon)q)T^2} \tag{9c}$$

with $\epsilon = \mathrm{M}_{H2O}/\mathrm{M}_{dry} \approx 0.622$ being the ratio of molar masses of water vapour and dry air. In LTT v1 (and v2), it is assumed
that refractivity decreases exponentially with an increase in height between adjacent vertical levels of an atmospheric model.
In the LTT ray tracer, coefficients are set in accordance with Bevis et al. (1994): $k_1 = 77.60$ K hPa$^{-1}$, $k_2 = 70.4$ K hPa$^{-1}$
and $k_3 = 3.739 \cdot 10^5$ K$^2$ hPa$^{-1}$. The RADIATE ray tracer uses coefficients by Rüeger (2002) with $k_1 = 77.695$ K hPa$^{-1}$,

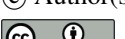



$k_2 = 71.295$ K hPa$^{-1}$ and $k_3 = 3.754 \cdot 10^5$ K$^2$ hPa$^{-1}$ (Landskron, 2018). The differences in slant delays due to different

coefficients are rather small, being about 1 mm at $5°$ elevation angle (Nafisi et al., 2012). Microwave refractivity increases slightly due to the presence of atmospheric particulate matter, such as hydrometeors, dust, and biogenic aerosols (Solheim et al., 1999; Navarro Trastoy et al.). However, in this study, their impact in slant delay computation is ignored.

## 3  Methodology

### 3.1  Improvements contained in LTT v2

Before discussing physical assumptions of the new algorithm, the key concepts underpinning the LTT v2 ray tracer are reviewed. The domain of the LTT v2 ray tracer solver (or, the LTT domain grid), is a two-dimensional grid of refractivity $N(i,j)$ and radius $r(i,j)$. The first index $i$ corresponds to the vertical model level of the NWP model, and $j$ the position of a vertical profile (more conveniently, a "column"). Thus, $i$ and $j$ are the vertical and horizontal coordinates in the LTT domain grid. Additionally, the station height, satellite position and top-of-atmosphere conditions are specified. This arrangement is illustrated

in Fig. 1.

The LTT domain is constructed by interpolating refractivity values from the regular NWP model grid onto the propagation plane, which intersects the Earth at a great circle defined by the station location and azimuth. The interpolated columns are equally separated by angular separation $\Delta\psi$ which equals to a longitude increment $\Delta\lambda$ of the regular Gaussian grid in the NWP model. As a default configuration, the LTT algorithm represents a propagation plane with 60 columns for NWP data

resolution 640x1280 ($\Delta\lambda = 0.28125°$), and 120 columns for 1280x2650 ($\Delta\lambda = 0.140625°$), which can be changed, if needed. Hence, the default limit of the great circle distance from the receiver to the point where the signal enters the atmosphere is around 1900 km and the LTT ray tracer is capable of computing tropospheric delay for as low as around $2°$ elevation angle signals. The contribution of slant delay from above the NWP model top is modelled with the Saastamoinen formula, and the ray is a straight line (Saastamoinen, 1972).



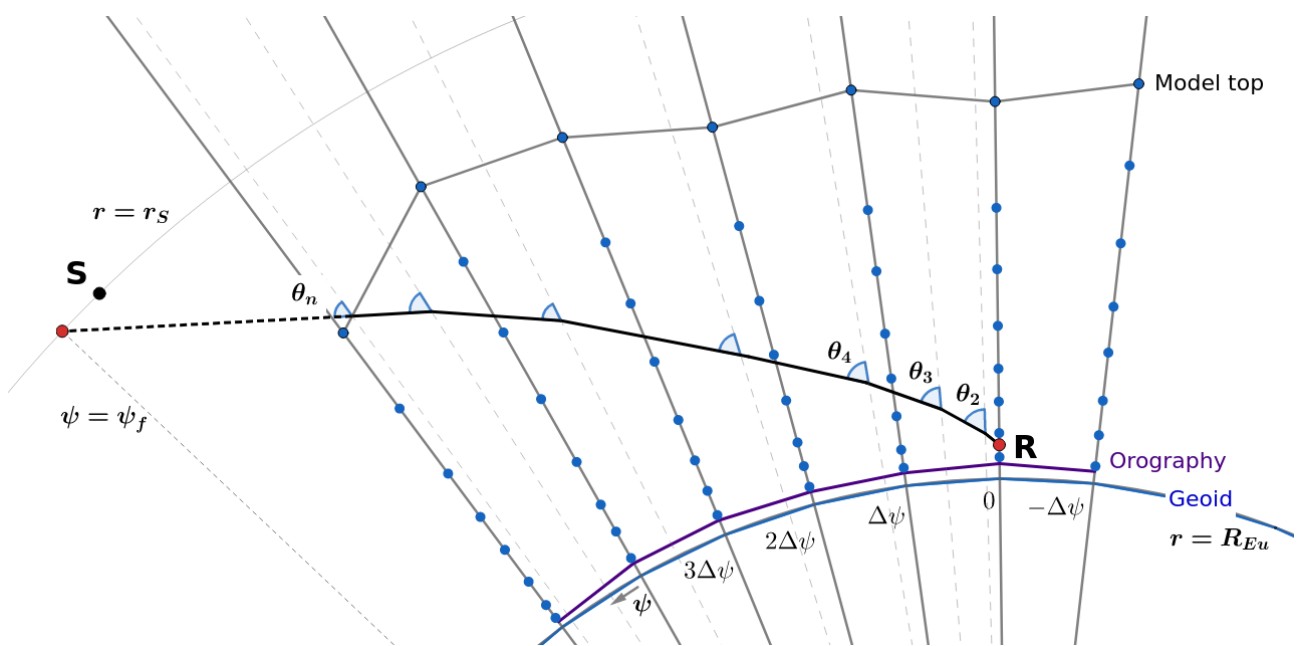

**Figure 1.** Illustration of the key concepts in the LTT algorithm. The RT integration starts at the ground station R and terminates at the radius $r_S$ (red points). The satellite is located at S (black point). The signal path estimate is denoted by black solid curve inside the model domain, while propagation is linear above the model atmosphere (dashed black line). Atmospheric refractivity is computed at the LTT domain grid points (blue dots), spanning from the ground (Orography, purple solid) to the highest model level of the NWP model (Model top, grey solid). Vertical coordinates of the ray and the LTT domain are bonded to the idealized geoid (Geoid, blue solid curve).

195 The domain's radial separation is in-homogeneous. The radius values of the ground station, the signal path and the domain grid are bonded to the selected coordinate system. In the LTT approach (both in LTT v1 and LTT v2), the shape of the geoid in the vicinity of the ground station is approximated with a sphere of a radius of curvature $R_{\mathrm{Eu}}$ which is the Euler radius of curvature in the WGS-84 reference ellipsoid (Rodgers, 2000). This way, an origin of polar coordinates for Equations 5 is the centre of curvature at the station position. The radial coordinates of the domain's grid points are computed separately for each

200 vertical profile as the model level height plus $R_{\mathrm{Eu}}$, as illustrated in Fig. 1. Model level heights are computed by solving the hydrostatic equation (Eqs. 13), as explained in Section 3.2. Station height is provided to the LTT domain as the mean sea level height. It is pre-computed from ellipsoid height with Eq. 8b using undulation products of Earth's geopotential model EGM2008 (Pavlis et al., 2012).

 The RT equations are restored to exactly follow the theory of Rodgers (2000). Hence the horizontal refractivity gradient

205 term has now been included in Eq. 5c. The along-plane horizontal gradient of a refractive index $\frac{\partial n(r,\psi)}{\partial \psi}$ at any location $(r,\psi)$ is computed under the assumption of refractivity changing exponentially in the vertical direction and linearly in the horizontal



direction in the sub-grid scale, as follows:

$$\frac{\partial n(r,\psi)}{\partial \psi} = 10^{-6}\left(\hat{N}_{j+1}(r) - \hat{N}_j(r)\right)/\Delta\psi \tag{10a}$$

$$\hat{N}_j(r) = N(i,j)\cdot\exp\left(-\ln\frac{N(i,j)}{N(i+1,j)}\cdot\frac{r-r(i,j)}{r(i+1,j)-r(i,j)}\right) \tag{10b}$$

where $N(.,.)$ and $r(.,.)$ are the known discretized values of refractivity and radius, such that point $(r,\psi)$ is inside the grid cell $(i,j),(i+1,j),(i+1,j+1),(i,j+1)$. $\hat{N}_j(r)$ is the interpolated refractivity value for the vertical profile $j$ at radius $r$. $\Delta\psi$ is the constant horizontal separation between columns. Horizontal gradient in the ray tracer has a minor impact on slant delay, altering the $5°$ elevation slant delay by less than $1\,\mathrm{mm}$. Therefore, this term can be neglected in the operational setup to reduce computational cost.

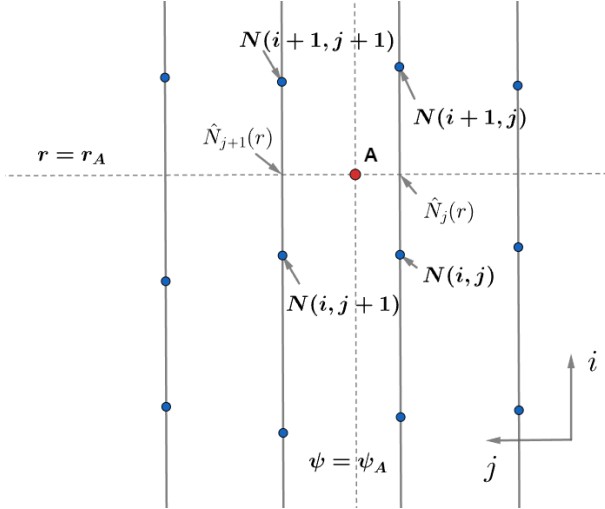

**Figure 2.** Illustration of the horizontal gradient computation at point A located at $(r,\psi)$. The adjacent refractivity values (N) needed in the formula are indicated in boldface. The intermediate values $\hat{N}(r)$ are vertically interpolated refractivities, and the gradient is computed by Eq. 10a. The blue dots indicate the LTT domain grid.

The geometrical delay term in Eq. 6 is included in the new LTT v2 algorithm. Direct estimation of a path length integral in Eq. 6 is prone to large numerical errors. Therefore, the extension of delay due to bending (i.e., geometrical delay) is calculated based on simple geometrical assumptions, which are illustrated in Fig. 3a. The geometrical delay $\Delta$ is the difference between the straight and the real path, approximated with the following formula:

$$\Delta = \int_S \mathrm{d}s - G \approx \frac{1}{2}R_{ToA}(\psi_{E'} - \psi_E)\cdot\delta\theta\cdot\cos\theta_E \tag{11}$$

with $R_{ToA}$ being the radius at the top of atmosphere, $\psi_{E'}$ and $\psi_E$ the $\psi$-coordinates of $E'$ and $E$, $\delta\theta$ the difference between the geometrical and real zenith angles at the receiver, and $\theta_E$ the ray's zenith angle at $E$. In Fig. 3a, $\triangle TEE'$ is much larger than $\triangle REE'$, and the atmospheric part of the signal $(RE)$ is assumed a straight line. Hence, the geometrical delay is $RE +$

$ET - RT$. Since the atmospheric part of the signal ($RE$) is assumed a straight line (Fig. 3a), the geometrical delay can be refined by calculating the length of the $RE$ curve conventionally. This refinement, however, is not implemented in the LTT v2.

The impact of geometrical delay $\Delta$ is seen at $20°$ elevation and below, reaching $50\,\mathrm{cm}$ at $5°$ elevation. At the same time, the value of $\Delta$ can vary significantly at low elevations, as this approximation is sensitive to the station height and orography. In Fig. 3b, $\Delta$ less than $10\,\mathrm{cm}$ for $5°$ elevation occurs only for stations POL2 and UNSA (IGS, 2024b, c), which both have steep orography in some azimuth directions.

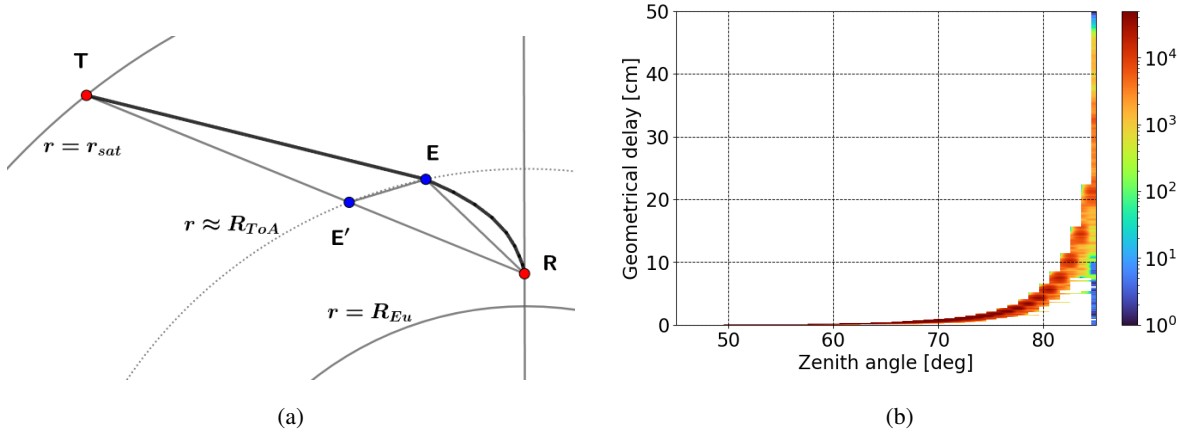

(a)

(b)

**Figure 3.** a) An idealized setup for geometrical delay calculation. The solid bold black curve is the signal path from a transmitter $T$ to the receiver $R$, entering the atmosphere at $E$. A straight signal path would reach the atmosphere at $E'$. The geometrical delay is $RE + ET - RT$. b) Two-dimensional histogram of geometrical delays. The geometrical delays are computed for 256 stations at 72 azimuth angles and 85 zenith angles on 5[th] of December 2016. The total number of values is around $3.8 \cdot 10^7$. Colourbar indicates the number of occurrences of geometrical delay $\Delta$ in each bin (40 equal zenith angle bins and 500 equal delay bins).

In LTT v2, the initial value problem of finding the zenith angle of a ray when it reaches the receiver (or, initial zenith angle, 230 $\theta_0$) is solved iteratively. Similarly to Eq. 7, the correction of initial zenith angle estimate $\theta_0^n$ yields the next value $\theta_0^{n+1}$:

$$\theta_0^{n+1} = \theta_0^n - \alpha \Delta\psi^n \tag{12}$$

with $\alpha$ being an arbitrary scaling parameter, $0 < \alpha \leq 1$, and $\Delta\psi^n$ the deviation of angle $\psi$ from the target value at the n-th iteration.

Figure 4 provides an example of an initial value iterative search in a selected LTT domain (see Fig. 1) for a variety of satellite 235 zenith angles. The initial-end conditions diagram is on the left plot and produced slant delays are on the right, computed for the first ten iterations by Eq. 12. As seen in Fig. 4 (left), the initial zenith angle adjustment is increasingly larger the larger satellite zenith angle is. Curiously, the convergence of the deviation $\Delta\psi^n$ is almost perfectly exponential which implies that the uncertainty of initial $\theta_0$ is proportional to the error of the end condition. Also, after many iterations, the deviation $\Delta\psi$ weakly depends on satellite zenith angle, in this example, reaching around $10^{-5}$ degrees by iteration #9 for all zenith angle values. 240 The slant delay value is adjusted throughout the iterative process and converges after 5-6 iterations, as shown in Fig. 4 (right).



In the LTT v2 algorithm, the number of iterations to produce the final delay value is fixed to 8 and $\alpha = 1/\sqrt{2}$, no threshold criteria on convergence is applied.

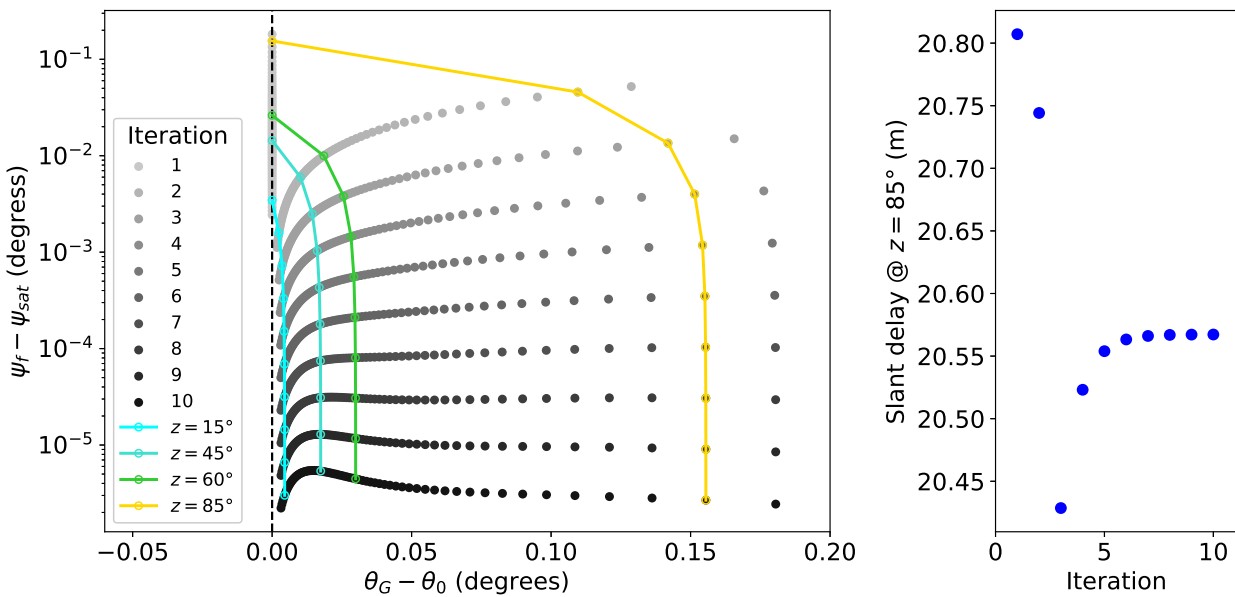

**Figure 4.** Difference between the geometrical zenith angle of the satellite ($\theta_G$) and the ray zenith angle at the receiver ($\theta_0$) versus the angular separation between the endpoint of the ray ($\psi_f$) and the satellite location ($\psi_{sat}$) for the first 10 iterations. The values in grey shades are plotted for different satellite zenith angles, ranging from $10°$ to $86°$ with $1°$ step. The coloured lines indicate the convergence trajectory for selected zenith angles (left panel). The resulting delay values at $5°$ elevation produced at each iteration (right panel). Scaling parameter $\alpha$ is $1/\sqrt{2}$. Data is produced using the LTT v2 algorithm for GNSS station JIUFENG (JFNG) on $5^{\text{th}}$ of December 2016, 07:00 UTC for azimuth angle $45°$.

## 3.2 Data input from the OpenIFS weather model

In this paper, all three RT algorithms (LTT v1, LTT v2, and RADIATE) are supplied with numerical weather data of the
OpenIFS weather prediction model, which is the atmospheric component of the Integrated Forecasting System (IFS) of the European Centre for Medium-Range Weather Forecasts (ECMWF) (ECMWF, 2017a). OpenIFS is a portable version of IFS intended for academic use. It produces forecasts with a skill identical to the full IFS. However, data assimilation is not included in OpenIFS, and the initial conditions must be supplied separately with the optimal ECMWF state estimates (Lean et al., 2021; Rabier et al., 2000). The version of OpenIFS used here is 43r3v1, which was part of the operational forecasting system at
ECMWF from July 2017 to June 2018.

The forecast setup is as follows. 12-hour forecasts are produced using initial states at 0000 and 1200 UTC for each day during December 2016. The model's time integration proceeds with 10 minutes step, and the output interval of the state





is 1 hour resulting in 744 atmospheric states per month. The output contains the following atmospheric variables: surface geopotential, logarithm of surface pressure, as well as temperature and relative humidity at model levels. The atmospheric

simulations are performed at a resolution of $T_L 1279$, corresponding roughly to 18 km horizontal resolution at the equator, at 137 vertical levels (+ a surface level) (ECMWF, 2022). Administration of the atmospheric simulations and output data is facilitated by the OpenEPS workflow manager (Ollinaho et al., 2021; Ollinaho, 2020).

The weather data is output at model levels, the heights of which are computed by the LTT program. In OpenIFS (ECMWF, 2017b), the construction of model-level orthometric heights $\mathbf{h}$ starts by defining the height of the surface level $\mathbf{h_{N+1}}$ using the

surface geopotential. The lowest model level $\mathbf{h_N}$ follows the model surface 10 m above it. The heights of model levels $\mathbf{h_i}$ are computed from the lowest model level $\mathbf{N}$ to the top of the atmosphere (level 1), as follows:

$$\mathbf{h_{N+1}} = \frac{z}{g} \tag{13a}$$

$$\mathbf{h_N} = \mathbf{h_{N+1}} + 10\text{m} \tag{13b}$$

$$\mathbf{h_i} = \mathbf{h_{i+1}} + \frac{1}{g} R_d T_{1/2}^* \ln(\frac{p_{i+1}}{p_i}) \tag{13c}$$

$$p_i = \frac{a_i + a_{i+1}}{2} + \frac{b_i + b_{i+1}}{2} p_s \tag{13d}$$

where, $z$ is the surface geopotential (in $\text{m}^2\,\text{s}^{-2}$), $g$ the constant mean sea level gravity ($9.80263\,\text{m}\,\text{s}^{-2}$), $R_d$ the dry air constant ($287.04\,\text{J}\,\text{K}^{-1}\,\text{kg}^{-1}$), $T_{1/2}^*$ (K) the half-layer virtual temperature, $p_i$ (Pa) the half-level pressure at level $i$ computed from the surface pressure $p_s$ using the sigma coefficients $a$ and $b$. Before vertical column construction, the atmospheric variables are bi-linearly interpolated from the horizontal grid.

### 3.3 OpenIFS input data to the RADIATE ray tracer

To deploy RADIATE with OpenIFS data, conversion to a format compliant with RADIATE's rigid requirements is needed. RADIATE requires weather data in a hardcoded format, mandating a spatial representation of data in a regular grid from 90 to $-90$ degrees in latitude and 0 to 359 degrees in longitude. The horizontal resolution of weather data is scaled to comply with default $1 \times 1$ degrees of RADIATE input. Despite being technically feasible to interpolate weather data with more fine

resolution, Hofmeister et al. (2015) found no strong evidence to suggest changing the default resolution. The vertical resolution must adhere to the 25 pressure levels used in ECMWF operational forecasts starting from 1 hPa downwards. The input file for RADIATE is a plain text file, featuring a hardcoded section containing information about resolution, extent, the list of pressure levels, the number of parameters, and the date of weather data, followed by the actual data - geopotential height Z (m), specific humidity q, and temperature T (K).

### 3.4 Production of slant delays

Information about the signal tropospheric slant delay is required in GNSS data analysis. To employ the modelled delays in GNSS Precise orbit determination (POD), we utilize the direct slant delay coupling approach of Navarro Trastoy et al. (2022), where the delay estimates are provided to the POD solver as-is, omitting the use of low-order parametrization by e.g. Vienna





Mapping Functions (Landskron and Böhm, 2018). The three ray tracers (LTT v1, LTT v2, and RADIATE) are configured to produce these products.

The slant delay products are as follows: slant delay values are computed in a regular azimuth-elevation grid for each station. These are called skyviews. Their resolution is $5 \times 1$ degrees in azimuth and elevation, respectively. Technically, the ray tracer algorithm is launched multiple times with a virtual GNSS satellite located at the grid points of a skyview. The delay estimate for an individual GNSS observation is obtained by spatially interpolating the skyview value for the azimuth/elevation angles, and temporally between the output epochs.

The RADIATE skyviews are produced as follows. The user is required to create an AZEL (azimuth-elevation) file, detailing the directions of observations. We are interested in a regular grid of azimuths and elevations, thus we use the option of creating a uniform AZEL file. The file format is specified in a separate file, where elevations and the number of uniformly distributed azimuths (set at 72 in our case) are defined.

The frequency of the slant delay products is 4 times per day (every 6 hours). The LTT v1 and LTT v2 algorithms digest slant delays with any time resolution, e.g. the hourly OpenIFS model output. However, we are limited by the design of the RADIATE software, which allows the desired number of output epochs per day set to a maximum of 4. Hence, 6-hour interval is selected to ensure a fair comparison.

### 3.5 LTT v2 as a software

We assembled the LTT v2 model into an openly available Fortran-language software with a modular structure. The weather data is uploaded to LTT v2 from a GRIB (GRIdded Binary) format file using the ecCodes (Najm, 2021) routines. Other required inputs are the station list containing a 4-character identifier, position and height for each station, as well as the IFS sigma coefficients defining the model levels. The LTT v2 program is configurable via the so-called namelist files, which specify the format of delays products and station height definition.

The core of the new ray tracer is the refined LTT operator routine, which maps the LTT domain, the satellite and the station coordinates (Fig. 1) to the slant delay value via solving the Eqs. 5. They are integrated as follows. Each NWP model level is divided into equal differential steps, and the signal path increments are computed in each step via a four-degree Runge-Kutta scheme. The number of steps per level and the number of initial zenith angle iterations are adjustable in the program. The baseline computing setup has 10 steps per level and 8 initial zenith angle iterations. A single baseline LTT RT execution costs $1-10$ CPU ms, depending on the station height and satellite elevation, and the 72x85 skyview takes $40$ CPU s to compute.

The LTT v2 program is designed to produce slant delay skyviews, however, other modelling strategies are possible. A viable option is to perform computation per observation, given the location, time, azimuth and elevation of the event. Thus, no interpolation is needed from a skyview value, but time interpolation remains. The main issue of this strategy is increased computational overhead due to the construction of the LTT domain for each observation. The GNSS network produces tens of millions of observations per day, which makes this approach unrealistic. However, geodetic systems with higher accuracy demands and fewer observations can benefit from per-observation slant delay computation.





## 4 GNSS processing and performance metrics

Assessing the quality of ray tracing in a neutral gas atmosphere is complicated due to lack of absolute reference. The approach here is to apply a software to process GNSS constellations and ground station networks. This allows validation of the different

ray tracers by comparing resulting GNSS products with all other inputs remaining unchanged. GROOPS (Gravity Recovery Object Oriented Programming System) is used here to produce the GNSS products. GROOPS (Mayer-Guerr et al., 2021) is an open-source software toolkit for gravity recovery that also includes a package for processing GNSS constellations and ground station networks. In GROOPS, the 'GNSSProcessing' program is used, which solves orbits and ground network station locations simultaneously using least-squares adjustment.

Regarding the GNSS products, there is no absolute reference. To validate the GNSS processing result, we use two metrics: midnight discontinuities of satellite orbits for consecutive days, and precise point positioning error of ground stations. GNSS products are computed daily in observation batches of 24 hours. The daily products are thus independent of one another. The midnight discontinuity (MD) means the distance between the last point of one day's orbit and the first point of the following day (Navarro Trastoy et al., 2022; Massarweh et al., 2021). For the precise point positioning (PPP), the daily station positions of

the set of 205 IGS stations are computed and compared with a long time series reference position from IGS14 (Rebischung and Schmid, 2016). We compute root mean square (RMS) and standard deviation (SD) of the differences to the reference positions. RMS indicates how close the PPP solutions are to the reference IGS14 solutions, whereas SD indicates the consistency of the PPP solutions.

The GNSS POD experiments are performed producing both metrics using the slant delay estimates from three ray trac-

ers: RADIATE (Landskron, 2018), LTT v1 (Eresmaa et al., 2021); and LTT v2. Thirty days of GNSS data are processed in December 2016.

## 5 Results

### 5.1 Comparison of slant delay estimates

Tropospheric slant delay estimates by LTT v2 are compared against its predecessor, LTT v1, and the VieVS Ray Tracer

(RADIATE). Figure 5 summarises the differences between these models showing the residual statistics for the slant delays in skyviews produced during one month for the set of 205 ground stations. The red areas on these histograms (Fig. 5 correspond to difference values of the overwhelming majority of slant delays, while the green and blue areas show the extent of rare extreme discrepancies between the models. Since the delay values are computed for a variety of weather and geographical conditions, Fig. 5 can be seen as a probability analysis of the gross errors in slant delay models, in analogy to Monte Carlo simulations.

The difference between the two LTT model versions is smaller and more asymmetric than the one between LTT v2 and RADIATE. As seen in Fig. 5a, slant delays of LTT v2 are typically smaller than those produced with LTT v1 at high elevations, whereas below about 75° zenith angle the opposite is true. There are two effects that might contribute to this behaviour: the refinement of the initial conditions in LTT v2 leads to a preferably smaller starting zenith angle $\theta_0$ (Fig. 4), thus decreasing the





delay, whereas the addition of the geometrical delay term to LTT v2 increases the delay at low elevations (Fig. 3b). According
to Fig. 5b, slant delays of LTT v2 deviate much more from RADIATE-based values than from LTT v1. The difference is close
to a normal distribution with a preference for delays of RADIATE being slightly larger than those of LTT v2 at the most of
zenith angles. At below $10°$ elevation, however, LTT v2 usually produces larger delays.

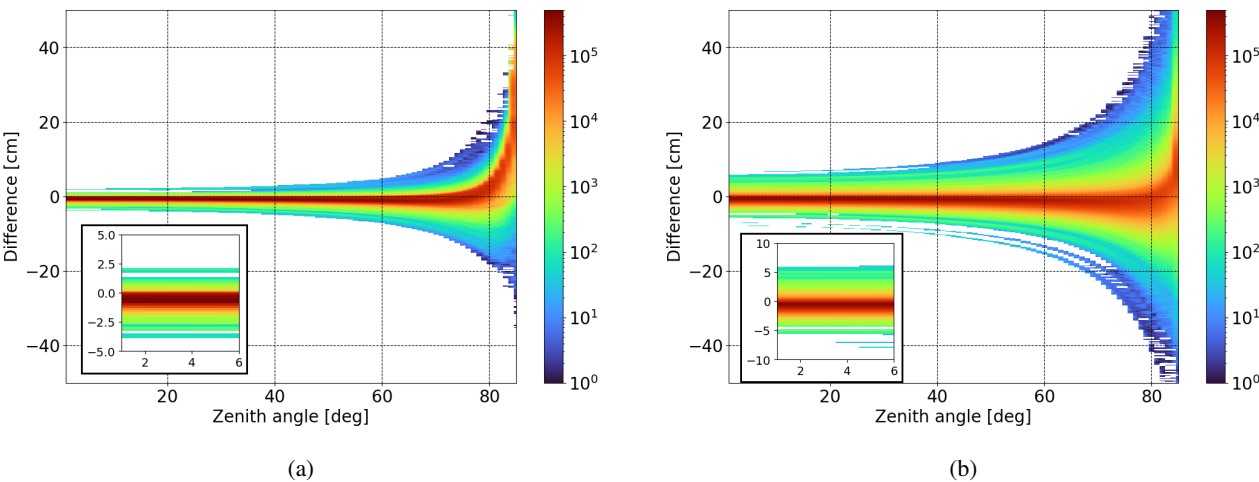

(a)                                                                  (b)

**Figure 5.** Two-dimensional histogram of ray-traced slant delays difference. (a) LTT v2 minus LTT v1, (b) LTT v2 minus RADIATE. The data
sample comprises around $1.9 \cdot 10^8$ slant delays (124 epochs at 256 stations and 72 azimuth angles by 85 zenith angles). Colourbar indicates
the number of occurrences of delay differences in each bin. There are 85 equal zenith angle bins and 500 equal delay bins

## 5.2 Comparison of GNSS products

The first metric used here is the midnight discontinuity (MD) for GPS satellites, specifically, the average positional MD of all
satellite orbits ($\Delta$) and the standard deviation related to this average ($\sigma_\Delta$),

$$\Delta = \frac{1}{N_{GPS}} \sum_{i=1}^{N_{GPS}} \sqrt{\delta r_{a,i}^2 + \delta r_{c,i}^2 + \delta r_{r,i}^2} \tag{14a}$$

$$\sigma_\Delta = \sqrt{\frac{1}{N_{GPS}} \sum_{i=1}^{N_{GPS}} (\sqrt{\delta r_{a,i}^2 + \delta r_{c,i}^2 + \delta r_{r,i}^2} - \Delta)^2} \tag{14b}$$

where, $N_{GPS}$ is the number of processed satellites in a GPS constellation (usually equal to 32), $(\delta r_a, \delta r_c, \delta r_r)$ the position
difference in along-track, cross-track and radial direction between the last point of one day's orbit and initial point of the
following day. The $\Delta$ and $\sigma_\Delta$ are computed daily for Dec 2016, resulting in 30 pairs of values. These values are shown in
Fig. 6, $\Delta$ in the left, and $\sigma_\Delta$ in the right. With these metrics, LTT v2 is an overall improvement over RADIATE showing more
consistent and accurate orbital products for 27 out of 30 days (Fig. 6, top left). The same conclusions are drawn when the
results are compared to LTT v1, yet the improvement is smaller in this case.



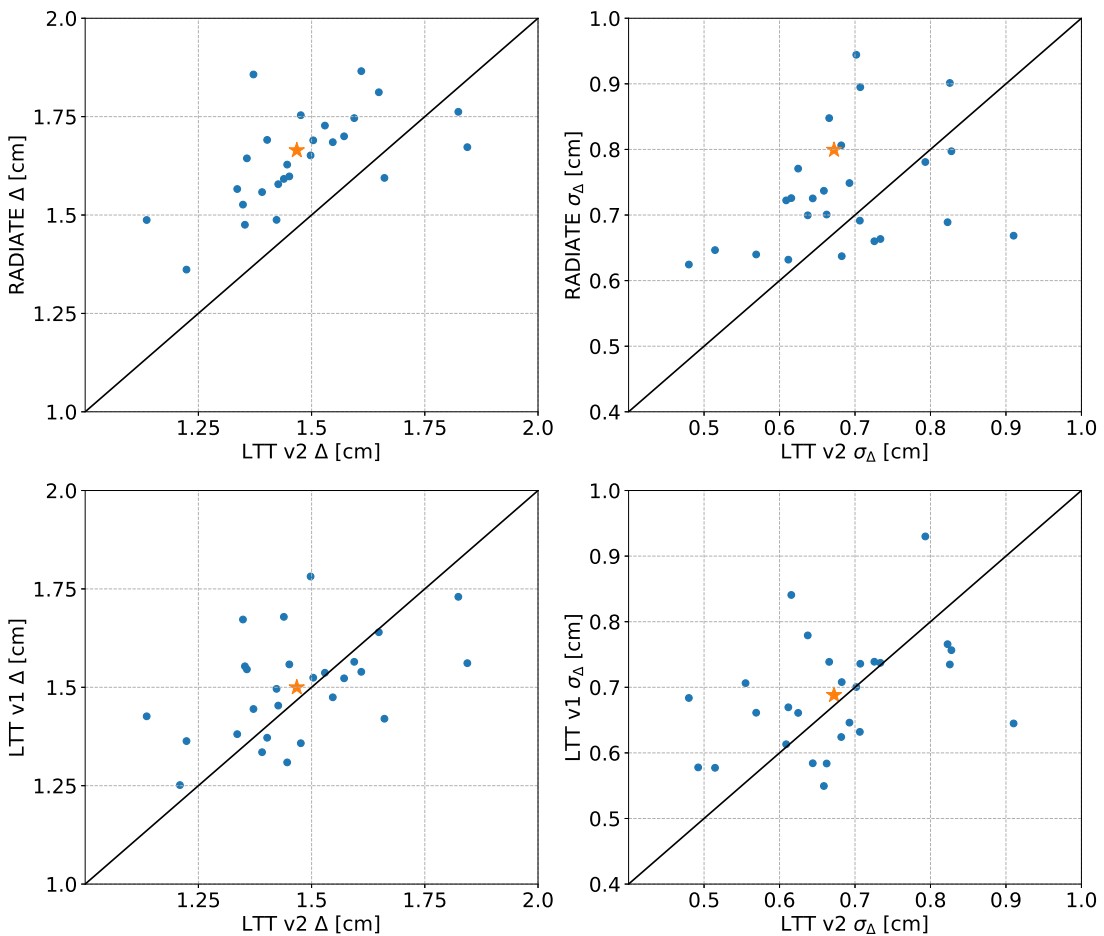

**Figure 6.** Comparison of daily midnight discontinuities ($\Delta$) and their standard deviations ($\sigma_\Delta$) with tropospheric slant delay estimates from different ray tracers for December 2016. Points are the mean values over the entire satellite constellation, computed with Eqs. 14. From top left clockwise: LTT v2 versus RADIATE ($\Delta$), LTT v2 versus RADIATE ($\sigma_\Delta$), LTT v2 versus LTT v1 ($\sigma_\Delta$), LTT v2 versus LTT v1 ($\Delta$). The star indicates the mean value, also reported in Table 1.

The second metric is related to the precise point positioning (PPP) of the ground stations involved in GNSS processing.

The position is defined by North, East and Height components. We analyse the residuals $x$ of daily positions of the set of 205 stations during 30 days in Dec 2016.

$$x = r - r_{IGS14} \tag{15}$$



with $r_{IGS14}$ being the reference daily position, and $r$ the result of the PPP processing. The positions are network solutions which comply with the net-zero rotation condition. In reality, the resulting reference frame is different each day due to noise and natural variations, however, position values are not transformed from this fluctuating frame towards the IGS14 reference frame. From the monthly time series of $x$, the metrics are computed for each ground station as follows:

$$x_{\mathbf{RMS}} = \sqrt{\frac{1}{30}\sum_{d=1}^{30} x_d{}^2} \tag{16a}$$

$$\sigma_x = \sqrt{\frac{1}{30}\sum_{d=1}^{30} (x_d - \overline{x})^2} \tag{16b}$$

where $\overline{x}$ is the mean residual, $x_{\mathbf{RMS}}$ the root-mean-square and $\sigma_x$ the standard deviation of residual in East, North and Height components. Squaring of vectors is performed element-wise. The distributions of $x_{\mathbf{RMS}}$ and $\sigma_x$ among different stations are shown in Figs. A1 and A2 of the Appendix, respectively. The mean values over all stations are shown in Table 1. It summarises the GNSS products comparison w.r.t. the four metrics: monthly average of midnight discontinuity and its standard deviation, the network average RMS and $\sigma$ of precise point positioning residuals. On average, the precise point positioning is significantly more reliable when using the LTT algorithms instead of RADIATE, both for RMS and $\sigma$ of position residuals. LTT v1 and LTT v2 demonstrate fairly similar skill, which implies that the differences in slant delay estimates induced by the model modifications provide a much smaller effect on PPP results than the improved use of weather model input data.

| | Monthly average $\Delta$, cm | Monthly average $\sigma_\Delta$, cm | Average $x_{\mathbf{RMS}}$, cm East, North, Height | Average $\sigma_x$, cm East, North, Height |
|---|---|---|---|---|
| **LTT v2** | 1.47 | 0.67 | 0.41 , 0.34 , 2.23 | 0.19 , 0.18 , 1.44 |
| **LTT v1** | 1.50 (+2.2%) | 0.69 (+2.3%) | 0.34 , 0.30 , 2.03 (−10.1%) | 0.19 , 0.18 , 1.43 (−0.6%) |
| **RADIATE** | 1.66 (+11.8%) | 0.80 (+15.9%) | 2.01 , 1.97 , 23.3 (+90.2%) | 0.69 , 0.78 , 5.26 (+72.7%) |

Table 1. The metrics for GNSS processing supplied with tropospheric slant delays from different ray tracers. Percentages show the improvement (positive) or deterioration (negative) by the LTT v2 algorithm when compared to this value, computed as follows: a *value* minus *LTT v2 value* divided by the *value*; average $x_{\mathbf{RMS}}$ and $\sigma_x$ are transformed into Euclidian distance. The green colour indicates LTT v2 being better in comparison, and the opposite is with the red colour.

## 6 Discussion

The three RT algorithms are principally similar. They employ a two-dimensional regime of ray propagation, discrete path bending, and continuous refractivity change being linear in horizontal and exponential in vertical directions. The weather forecast data is the same for these RT algorithms. The delay estimates by the three algorithms are provided in the same format and resolution to the verification procedure of obtaining global navigation solutions. However, verification via the GNSS processing demonstrates different performances of these three RT models. Many details seem to be important.





First, the atmospheric state is input differently in RADIATE and LTT. RADIATE employs global NWP data at 1x1 degree resolution at 25 hard-coded atmospheric pressure levels, which are interpolated from the full OpenIFS model resolution definition. LTT v1 and LTT v2, in contrast, access the original NWP data at the native resolution (around 0.14 x 0.14 degrees) at 137 model levels. Downscaling both the horizontal and vertical resolutions truncates small-scale atmospheric phenomena, leading to less informative delay in RADIATE than in LTT. It is worth noticing, that the time resolution of slant delays is 6 hours and time-linear interpolation is applied. This hinders the influence of time evolution and movement of weather patterns both in RADIATE and LTT delay estimates.

Second, the RT domain grid is created differently in the three RT algorithms. Both LTT versions employ a sequence of model level heights to build up the grid vertically, while RADIATE has a different setup. LTT v1 constructs full-pressure model level heights, while LTT v2 makes use of half-level pressure values. In addition, the gravity acceleration $g$ in the hydrostatic equation (Eq. 13) is changed to a constant value in LTT v2. While not physically correct, height-constant $g$ is technically consistent with the OpenIFS model. It is important to note, that the LTT v1 model is a modification of the slant delay operator by Eresmaa et al. (2008). The latter was intended as an extension of the High-Resolution Limited Area Model (HIRLAM), whose model level definition is different from that of OpenIFS.

Although the main physical assumptions in the three RT algorithms are the same, the details in the implementation are different. In LTT v2 we include plausible physical improvements. Attribution of impacts of different improvements is not performed in full since they influence each other in the GNSS processing. Considering the importance of correctly utilizing the weather model data, improving the physical assumptions is in somewhat minor role. We emphasize that LTT v2 has a principal advantage over LTT v1 and RADIATE models since it is directly compatible with the Integrated Forecasting System (IFS) at any present or future resolution.

## 7 Conclusions

We have developed a signal delay model which is closely tied to the OpenIFS numerical weather model ensuring nearly lossless representation of the atmospheric state for solving a signal propagation. The LTT v2 algorithm has a more robust mathematical foundation of signal propagation than the previous version (LTT v1). Our signal delay model validation via GNSS processing concludes that LTT is superior to RADIATE, and we attribute this mainly to the limitation of atmospheric data representation in RADIATE. Two LTT models, on the other hand, demonstrate a fairly similar skill, suggesting that the physical scheme improvements of RT have less impact on slant delay quality than the proper adaptation of numerical weather data into RT. In comparison with earlier developments in the atmospheric signal delay of GNSS and other geodetic systems, the main innovation here is the relaxation of all assumptions regarding the use of weather model data. It is now possible to fully enjoy the increasing quality and skill of weather model data.

From the perspective of the future of RT applications, the newly developed LTT v2 is well positioned for the emerging next-generation high-resolution global models (nextGEMS, 2024; ECMWF, 2024a; Hohenegger et al., 2023; Rackow et al., 2024). Since LTT v2 can ingest atmospheric model output data at any horizontal and vertical resolution, only a minimal amount of



post-processing of the model fields is required. This means that LTT v2 is ready to take full advantage of the fine-scale details of meteorological information provided by the next generation high-resolution models.

*Code and data availability.* The Least travel time model version Two (LTT v2) is available at Vasiuta (2024). License: GPL-3.0.

The Least travel time algorithm version One (LTT v1) is available at Eresmaa et al. (2021). License: Creative Commons Attribution 4.0

International licence.

The VieVS Ray Tracer software (RADIATE) is available at Landskron (2018). License: GPL-3.0.

The GROOPS toolkit software (Release 2021-02-02) is available at Mayer-Gürr (2021). License: GPL-3.0. Modification to GROOPS enabling usage of skyview tropospheric delay representation is available at Navarro Trastoy (2024).

The ecCodes package (Release 2.22.1) is available at Najm (2021). License: Apache-2.0.

The OpenIFS 43r3 model is available at ECMWF (2017a). A user licence is required.

The Open Ensemble Prediction System (OpenEPS) (Ollinaho, 2020) is used to launch the OpenIFS weather forecasts. License: Apache-2.0.

The pre-processing data for GNSS analysis is available at Strasser and Mayer-Gürr (2021). License: Creative Commons Zero v1.0. Universal.

The GPS orbits and observations are available at IGS (2024a). An example of the processing configuration is available at Navarro Trastoy (2024).

The initial atmospheric states used by OpenIFS are obtained from the OpenIFS Data Hub (ECMWF, 2024b). Data acquisition requires an OpenIFS user licence. The parameters for the data request are described in Section 3.2.

*Author contributions.* According to CRediT author statement [1]:

Conceptualization: HJ

Data curation: MV, LT

Formal Analysis: MV, ANT

Funding acquisition: HJ

Investigation: HJ, MV, ANT

Methodology: MV

Project administration: HJ, MV

Resources: HJ, LT

Software: MV, LT, ANT

Supervision: HJ

Validation:

Visualization: MV, ANT

Writing – original draft: MV, LT

Writing – review & editing: LT, ANT

---

[1]https://www.elsevier.com/researcher/author/policies-and-guidelines/credit-author-statement





*Competing interests.* Authors declare that they have no competing interests.

*Acknowledgements.* We gratefully acknowledge funding from the Academy of Finland (no. 1333034), the Finnish Academy of Science and Letters (Väisälä Fund), the Finnish Society of Science and Letters (Magnus Ehrnrooth Foundation), and the Maj and Tor Nessling Foundation
(no. 202200364). We gratefully acknowledge the super-computer resources provided by CSC - IT Center for Science in Finland. In particular, we are thankful for the technical support of Dr. Juha Lento from CSC.

## Appendix A: Statistics of station position residuals

The process of GNSS ground station positioning yields one residual estimate for one station per day, producing 205 time series with 6150 data points in total in our experiment. We performed three experiments: LTT v1, LTT v2 and RADIATE. The
distribution of the root mean square and the standard deviation of the residual time series are shown in this section.



## A1   Root mean square



**Figure A1.** Distribution of root mean square of position residuals $x_{\mathbf{RMS}}$ for IGS 205 ground stations. In the three-by-three table of figures, columns distinguish between the components of $x_{\mathbf{RMS}}$, and rows distinguish between different slant delay products.



## A2 Standard deviation

**Figure A2.** Distribution of standart deviation of position residuals $\sigma_x$ for IGS 205 ground stations. In the three-by-three table of figures, columns distinguish between the components of $\sigma_x$, and rows distinguish between different slant delay products.



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
