# Peer review of "Least travel time ray tracer, version Two (LTT v2) adapted to the grid geometry of the OpenIFS atmospheric model"

_Geoscientific Model Development, 2024_

## Author Comment (AC1)

Dear reviewer,

Thank you for your valuable comments. The commentary concerns the evaluation of the ray tracing delay models via GNSS data processing. In the reply to you, we are giving a thorough discussion and minor additions to the manuscript text.

**1.** The result of GNSS data processing is position and velocity states, as well as clock parameters, ionospheric electron content, integer ambiguities and phase biases. We have selected two types of metrics that represent the stability of the states: one being related to satellite orbits, the second related to station positions. The discussion by Strasser et. al. (2019) suggests that clock parameters are stable over daily processing time, which makes them less informative to analyse. Also, a subjective choice was made to ignore all parameters but orbits and station position, as was done by Zus et. al. (2021) in a similar experiment to analyse atmospheric mismodelling effects. We are adding extra context to the potential readers explaining our choice.

Line 325 REMOVE

Regarding the GNSS products, there is no absolute reference. To validate the GNSS processing result, we use two metrics:

Line 325 INSERT

The products of GNSS data processing are position and velocity states, as well as clock parameters, ionospheric electron content, integer ambiguities and phase biases. Since there is no absolute reference, the processing is evaluated by measuring the stability of the solution. Discussions by Strasser et. al. (2019) and Zus et. al. (2021) suggest that among all the most informative are orbit and ground station positions. Hence, we choose two metrics to validate ray tracing models:

**2.** The report of statistics in the Table 1 is correct. Indeed, the high average value of station height RMS and standard deviation for RADIATE is due to several huge values. The histograms at Figure A1 and A2 are constructed so that values higher than 10 cm are not visible in the figure. We decided not to remove these points as statistical outliers, because the same stations mismodelled in the RADIATE experiment are processed adequately in the LTT v1 and v2 experiments. This notion is hereby explained in the text as following:

Line 378 REMOVE

On average, the precise point positioning is significantly more reliable when using the LTT algorithms instead of RADIATE, both for RMS and σ of position residuals.

Line 378 INSERT

PPP residuals are generally smaller when using the LTT algorithms compared to RADIATE. In the RADIATE experiment, the height estimates are very inconsistent for some stations, which leads to high average RMS and σ values. These high values do not appear in Figs. A1 and A2.

**3.** In the text, we mention that LTT v1 and v2 models demonstrate fairly similar skill. We are not able to draw deeper conclusions. The approach of the processing is to estimate many parameters in the same least squares adjustment. Hence, orbit and station position states are entangled in the fitted solution. The orbit discontinuities and station position offsets (and behavior of other parameters) are indicative of stability of the solution as a whole. To quantify that would require a unified metric, to develop which would be a good idea for future work.
Another notion is that between separate experiments the percentage change of orbit metric is lower than for station metric (coloured numbers in Table 1). One might speculate that the station metric is more sensitive to mismodelling than the orbit metric. Again, future sensitivity tests are needed to prove that.
Small change to the text:

Lines 379-381 REMOVE

LTT v1 and LTT v2 demonstrate fairly similar skill, which implies that the differences in slant delay estimates induced by the model modifications provide a much smaller effect on PPP results than the improved use of weather model input data.

Line 379 INSERT

LTT v1 and LTT v2 demonstrate comparable skill, with v1 being better at precise point positioning and v2 better at orbit determination. This implies that modifications in ray tracing models from the old to the new version have very minor effect compared to improved use of weather model data.

**4.** We decided to keep the formula unchanged, since the values of standard deviation are used in comparative manner inside the paper. And, they are not shown against external processing results, such as by operational analysis centers.

References:

Zus, F., Balidakis, K., Dick, G., Wilgan, K., and Wickert, J.: Impact of Tropospheric Mismodelling in GNSS Precise Point Positioning: A Simulation Study Utilizing Ray-Traced Tropospheric Delays from a High-Resolution NWM, Remote Sensing, 13, https://doi.org/10.3390/rs13193944, 2021.

Strasser, S., Mayer-Gürr, T., and Zehentner, N.: Processing of GNSS constellations and ground station networks using the raw observation approach, Journal of Geodesy, 93, 1045–1057, https://doi.org/10.1007/s00190-018-1223-2, 2019.

---

## Author Response (AR1)

Dear Editor,

We have responded to two referees during discussion stage. The author's replies are uploaded as open comments in the Interactive Discussion. The total changes to the manuscript are listed here. Line numbers correspond to revised version of the manuscript.

1. We have slightly modified abstract to be in better agreement with the comments from the referee two.

2. The reported metrics values have been slightly changed in the abstract, since we have updated the results, as explained to reply to the referee two.

3. We have slightly modified introduction, to be in better agreement with the comments from the referee one and two:
3.1 The motivation of the work better explained: lines 40-53
3.2 The modelling setup better explained: lines 54-60
3.3 Added details about metrics of GNSS products: lines 62-63

4. We have removed reference to unpunished work: line 180

5. Changes to GNSS data processing explanation, to fulfill the comment of the referee one: lines 329-340

6. Section 5 (Results) reworked to be in better agreement with the comments from the referee two. The results of LTT-based models and experimentation with the RADIATE model are reported separately. This changes the subsections of Section 5. We also added more explanations throughout the text to be in better agreement with the comments from the referee one.

7. Related to change of Section 5, Table 1 have been split into two parts, and extra hints have been added for improved readability. This reflects the suggestions from the referee two.  The reported metrics values have been slightly changed in tables.

8. Conclusions have been partly modified to be in better agreement with the comments from both referees: lines 428-438.

* Several minor grammatical and cohesion fixes have been made throughout the text.